# Solving the B-SAT Problem Using Quantum Computing: Smaller Is Sometimes Better

**DOI:** 10.3390/e26100875

**Published:** 2024-10-18

**Authors:** Ahmad Bennakhi, Gregory T. Byrd, Paul Franzon

**Affiliations:** Department of Electrical and Computer Engineering, North Carolina State University, Raleigh, NC 27695, USA; gbyrd@ncsu.edu (G.T.B.); paulf@ncsu.edu (P.F.)

**Keywords:** quantum computing, B-SAT, Boolean satisfiability problem, Grover’s search, electronic design automation (EDA), conjunctive normal form (CNF), closed-box testing

## Abstract

This paper aims to outline the effectiveness of modern universal gate quantum computers when utilizing different configurations to solve the B-SAT (Boolean satisfiability) problem. The quantum computing experiments were performed using Grover’s search algorithm to find a valid solution. The experiments were performed under different variations to demonstrate their effects on the results. Changing the number of shots, qubit mapping, and using a different quantum processor were all among the experimental variables. The study also branched into a dedicated experiment highlighting a peculiar behavior that IBM quantum processors exhibit when running circuits with a certain number of shots.

## 1. Introduction

The application of quantum computing to electronic design automation (EDA) is an increasingly growing field that is gaining traction as quantum processors improve by the year [1,2]. Universal gate quantum processors are quite efficient at performing tasks that could be parallelized, and closed-box EDA testing has a lot of parallelization potential, since it needs to run all possible inputs to verify the function of the digital logical unit. One way to auto-construct a closed-box digital logic unit is to lay it out as a Boolean satisfiability (B-SAT) problem.

An early application of quantum computers was to solve the Boolean satisfiability problem using Grover’s search algorithm [3]. Grover’s search algorithm has long been hailed as one of the flagship algorithms of quantum computing [4]. Its ability to locate an item in an unstructured list has a complexity of $O(\sqrt{n})$. That being said, quantum computers are far from proving quantum advantage.

While we are still in the noisy intermediate-scale quantum (NISQ) era, various approaches from all angles are being leveraged to achieve the goal of quantum advantage. The depth of quantum circuits is usually used as an early indicator of how reliable the results will be. However, as demonstrated by the experiments in this study, you can still obtain both reliable results in quantum circuits with high depth, and unreliable results from quantum circuits with shallower depth.

## 2. Background

### 2.1. The Boolean SAT Problem

In this study, we try different adjustments to various configurations of B-SAT circuits. The goal was to note which configurations would yield the best results. The experiments are particularly useful for scientists who are planning on using quantum computing, Grover’s search algorithm to be precise, to solve either a Boolean satisfiability problem or a digital logic circuit that has been translated into conjunctive normal form (CNF) or even for closed-box testing a digital logic circuit. The variables included the number of shots, qubit mapping, and using a different quantum processor. The variables in the CNF circuit were the number of AND, OR, and NOT operators present. Here, we are talking about the logical operators and not the physical logical gates. The circuit was constructed in conjunctive normal form (CNF), which is one of the most widely used forms when structuring a Boolean satisfiability problem. A typical Boolean satisfiability equation can be expressed in the following way:(1)$$\begin{array}{cc} f\left(x_{1},\;x_{2}\ldots x_{n}\right)= & \left(x_{1}\vee \neg x_{2}\vee x_{3}\ldots \vee x_{n}\right)\\ & \wedge \;\left(\neg x_{1}\vee x_{2}\vee x_{3}\ldots \vee x_{n}\right)\\ & \;\wedge \ldots \left(x_{1}\vee \neg x_{2}\vee x_{3}\ldots \vee x_{n}\right) \end{array}$$

Each CNF equation is composed of clauses and literals. A literal is an instance of a variable inside a clause/parenthesis, which may or may not be negated. A clause is the OR of one or more literals grouped inside of a parenthesis. Finally, when clauses are ANDed together, they create a CNF equation. Not all variables have to be present in each clause, and this gives CNF great flexibility to represent a wide range of scenarios. Tseytin transformation [5] makes CNF an optimal form for combinational logic circuits to be converted into, since any circuit can be structured in CNF using De Morgan’s law.

A CNF circuit can be divided into three distinct stages, as seen in Figure 1. The first stage is the inversion of the inputs (blue box in Figure 1), or the creation of literals. As mentioned earlier, not all inputs have to be inverted. The varying degree of input inversion (per clause) is the main factor that would add complexity to the Boolean satisfiability problem and help us avoid duplicate clauses as much as possible. The following stage is inserting the literals in their respective OR gates to create a clause (red box in Figure 1). After that, all OR gate outputs are ANDed together to create the output of the Boolean satisfiability problem (purple box in Figure 1).

Assuming that *m* is the number of clauses and *n* is the number of variables in the Boolean satisfiability equation, a proportion parameter α=m/n is found to have major significance in terms of equation solvability. According to studies performed on the 3-SAT problem [6,7], there exists a critical point $\alpha_{c}\approx 4.267$ where the probability of finding a valid solution drops drastically for any number above it. This is why we made sure to keep α below this threshold for all experiments in our study.

### 2.2. Grover’s Search Algorithm

The 3-SAT problem has been at the forefront when it comes to exemplifying the applications of Grover’s algorithm. Grover’s algorithm is composed of three components:Hadamard initiationGrover oracleAmplitude amplification

Hadamard initiation is a common initialization stage seen in many quantum algorithms, where all qubits are flipped to superposition. This creates an equally weighted superposition of all computational basis states, thus harnessing one of the major perks of quantum computing. The next step would be the Grover oracle. The Grover oracle is a quantum circuit component that flips the phase of a state that satisfies a desired condition. In our experiment, the CNF equation would be translated into a Grover oracle. The last part of Grover’s search algorithm is the amplitude amplification stage. Unlike the Grover oracle, the amplitude amplification phase is not case- or result-dependent, meaning that the quantum circuit for the amplitude amplification remains identical for all cases and is always U=2|s〉〈s|−1.

After the Hadamard initiation, the algorithm works by repeatedly applying a Grover oracle and amplitude amplification for
$$\left\lfloor \frac{\pi }{4}\sqrt{\frac{N}{m}}\right\rfloor$$
where *N* is the number of entries on the list or $2^{n}$. *n* is equal to the number of qubits representing the variables in the Boolean satisfiability formula. *m* is the number of viable distinct solutions. Finally, the results are recorded by measuring all qubits. By exchanging the Grover oracle with a CNF circuit, we would be constructing a Boolean satisfiability solver. The layout can be seen in Figure 2.

### 2.3. Circuit Depth

Quantum circuit depth is a metric that is calculated based on the longest path between the data input and the output. It is a commonly used parameter to rate computational complexity and efficiency. In simple terms, deeper circuits imply more complex computations, but can also lead to increased susceptibility to errors due to longer interaction times between qubits and external influences. Qubit operations that can be performed in parallel do not accumulate but simply count as a single increment in the count (as seen in Figure 3).

There have been numerous algorithms designed to minimize quantum circuit depth, while still achieving the desired computational outcome. A shallow circuit depth is advantageous, as it reduces the potential for error accumulation, which is especially critical given the inherent fragility of quantum states. Additionally, shallower circuits can result in faster computation times, contributing to improved efficiency in quantum algorithms. While shallow circuits are desirable, reducing circuit depth might lead to an increase in the number of required qubits or gates in a quantum computation, potentially offsetting the benefits. Striking the right balance between circuit depth, gate complexity, and error correction is a non-trivial task in quantum algorithm development.

### 2.4. Shot Statistics

This section will mainly discuss the concept of shots in the classical and quantum computing context. “Shots”, in the context of quantum computing, play a crucial role in understanding the outcomes and statistics of quantum circuit executions. Shots are a fundamental parameter in quantum computing experimentation and are often used to describe the number of times a quantum circuit is run or measured at circuit termination.

In quantum mechanics, the act of measuring a qubit causes it to “collapse” into one of its possible states, resulting in a measurement of either a 0 or a 1. This collapse is probabilistic, meaning that the outcome of a measurement is not deterministic but is determined by the quantum state of the qubit. Consequently, the same quantum circuit executed multiple times may produce different measurements each time, due to the inherent uncertainty in quantum measurements. The number of these quantum measurements is often called “shots” in the experimental context. Each shot corresponds to one execution of the quantum circuit. By running a quantum circuit multiple times (with a specified number of shots), researchers and quantum computing programmers can collect statistical data related to the outcomes of the measurements.

The average mean of a classical operation outcome can be denoted by
(2)$$E\left[X\right]=\sum_{x\in 0,1}xPr\left[X=x\right]$$
where E[X] is the expectation value of the quantum circuit, while x is meant to represent the classical measurement outcome that would be multiplied by the probability of it happening: Pr[x]. This can be further reduced to Pr[X=1] and result in it being just *p*, which is just the probability of measuring 1. When translated into quantum computing terms, the average mean formula would look something like this:(3)$$\begin{array}{ccc} E[X] & = & \sum_{x\in 0,1}x\langle \hat{\mu }\left(x\right)\rangle \\ \; & = & \langle \sum_{x\in 0,1}x\hat{\mu }\left(x\right)\rangle \\ \; & = & \langle \hat{M}\rangle \end{array}$$

$\hat{\mu }$ is an observable. An observable is a measurable property of a circuit, where it could be a vector or a combination of different vector outcomes. Since the expectation value is a linear function, the constant *x* could be moved to the inside of the $\hat{\mu }$ bracket (as seen in Equation (Equation 3)). This would equal the quantum expectation value of the quantum observable, as demonstrated in line 3 of Equation (Equation 3). This $\hat{M}$ value is equal to the projector for the |1〉 vector, which symbolizes the probability of measuring a 1. In vector and matrix form it would equal this:(4)$$\begin{array}{ccc} \hat{M} & = & 0|0\rangle \langle 0|+1|1\rangle \langle 1|\\ \; & = & \left(\begin{array}{cc} 0 & 0\\ 0 & 1 \end{array}\right) \end{array}$$

The variance of a classical random variable (denoted by V), which describes the deviation around the mean, looks like the following equation:(5)$$\begin{array}{ccc} V[X] & = & E[{\left(X-E\left[X^{2}\right]\right)}^{2}]\\ \; & = & E\left[X^{2}\right]-E{\left[X\right]}^{2} \end{array}$$

This formula is shared by both the quantum and classical aspects, but where they differ is the way that the expectation value squared ($E[X^{2}]$) is reached. The difference is demonstrated in Table 1. At the end of the quantum path, we can see that the expectation value squared is equal to the observable operator matrix squared. By combining both sides of the variance equation, we can induct that
(6)$$\begin{array}{ccc} V[X] & = & \langle {\hat{M}}^{2}\rangle -{\langle \hat{M}\rangle }^{2}\\ \; & = & p-p^{2}\\ \; & = & p(1-p) \end{array}$$

The **chi-square**, denoted by $\chi^{2}$, is a statistical parameter that measures how close model data are when compared to observed data. It is formulated in the following equation:(7)$$\chi_{c}^{2}=\frac{{\left(O_{i}-E_{i}\right)}^{2}}{E_{i}}$$
where *O* represents the observed values, $E_{i}$ represents the expected values, and c denotes the degrees of freedom. The larger the disparity between the observed and model values, the larger the $\chi^{2}$ value. In other words, the smaller $\chi^{2}$ is, the better the model/experiment appears to be. More detailed information can be found regarding this subsection at [8].

#### Number of Shots

A single measurement (shot) of a quantum circuit has a probability of measuring a particular observable *x*, which would be written as
(8)$$\mathrm{Pr}\left[X=x\right]=\langle \hat{\mu }\left(x\right)\rangle$$

The average over all the random outcomes measured would be denoted as *S* and formulated in the following way:(9)$$\;S=\frac{1}{M}\sum_{m=1}^{M}X_{m}$$
where the first outcome is $X_{1}$ and the last measurement outcome is $X_{M}$. The number of shots measured here is *M*. Equation (Equation 6) is applicable in the context of the number of shots, and by combining it with Equation (Equation 5), we can induct that
(10)$$\;E\left[X_{m}\right]=E\left[X\right]=p\;\forall m\in \left\{1,...,M\right\}$$

As stated earlier, the expectation value is linear functional and is even utilized in Equation (Equation 5). This property can be also used to derive the distributed expectation value, as seen in the following equation:(11)$$\;E\left[aX_{m}+bX_{n}\right]=aE\left[X_{m}\right]+bE\left[X_{n}\right]\;\forall m,n\;a,b\in C$$

Again, when using Equation (Equation 5) to come up with the variance of a distributed random variable system (such as the quantum uncertainty), we can infer that the variance would be equal to
(12)$$\;V\left[aX_{m}+bX_{n}\right]=a^{2}E\left[X_{m}\right]+b^{2}E\left[X_{n}\right]\;\forall m,n\;a,b\in C$$

Following on from Equation (Equation 10) and combining it with Equation (Equation 9), we can assume that the variance of the mean for quantum systems can be laid out in the following way, given *M* number of shots is taken into account:(13)$$\begin{array}{ccc} V[S] & = & V[\frac{1}{M}\sum_{m=1}^{M}X_{m}]\\ \; & = & \frac{1}{M^{2}}\sum_{m=1}^{M}V\left[X\right]\\ \; & = & \frac{1}{M}V\left[X\right]\\ \; & = & \frac{p(1-p)}{M} \end{array}$$

This would indicate that, in an ideal quantum computing system, the more measurements or shots, the smaller the standard deviation and variance becomes. In other words, you can suppress quantum projection noise with enough shots. This can be demonstrated in the following example: Let us assume a 1-qubit system with a 50/50 percent chance of giving an outcome of either a 0 or 1 and that has been measured *M* number of times. The outcomes when M=2 are {0,0}, {1,0}, {0,1}, or {1,1}. The average of these sums would be repeated as displayed in Figure 4. The probability of two of those measurements would yield an average sum of 0.5({1,0} and {0,1}), while the other two possible measurements would give us average sums of 0({0,0}) and 1({1,1}). When *M* is increased, the variance and standard deviation become gradually more distinct, with outcomes that are distant from 0.5 becoming less probable to be measured (as in Figure 4).

## 3. Related Works

### 3.1. The Grover Search Approach

Utilizing Grover’s algorithm to solve the SAT problem is not something new, as there was a study that outlined its usefulness more than two decades ago [9]. The subject gained traction later when a couple of noteworthy studies were published [3,10]. While the earlier work [3] laid out the scaffolding of the bounds, it also asserted that “the classical part can simply be replaced by quantum hardware which does the same”. The other study [10] suggested a quantum–classical hybrid approach to solve the 3-SAT problem by offloading some of the calculations onto classical processors. It should be noted that even the qubits in the study were merely simulated, due to the technology of the times.

There has been a resurgence when it comes to studies using Grover’s search to solve and analyze satisfiability problems. A research paper [11] explored applying Grover’s search and constraint satisfiability to solve integer case constraints. The paper simulated and applied quantum circuits on Qiskit quantum simulators and IBM quantum processors, respectively. The authors also tried applying valid optimizations, while also discussing optimizations that would not yield better results. Among their applied optimizations were adding thermal relaxation and depolarization noises.

A recent study [12] outlining the effect of applying Grover’s search on different IBM quantum processor architectures showed interesting results. The different quantum architectures affected the results in unexpected ways, with the architecture that had the greatest number of qubits (ibmq_16_melbourne) not outperforming the other architectures, even though it outnumbered the others by over 10 qubits. There was an examination of the transpilation circuit result and why the results happened to turn out the way they did. The optimizations performed in the study were Qiskit library-driven optimizations rather than customized ones.

### 3.2. The Ising Model/QUBO Approach

Quadratic unconstrained binary optimization (QUBO) has been used in recent studies to solve Boolean satisfiability problems. The Ising model is closely related to the QUBO problem and is computationally equivalent. Most QUBO problems are converted into an Ising model, because it uses a Hamiltonian that translates well when run on a quantum annealing processor. The resurgence of such usage is due to the availability of commercial quantum annealer processors from D-Wave Systems [13]. Moreover, a myriad of problems can be formulated to fit into the QUBO model, as outlined in the following survey [14]. A detailed tutorial describing how a QUBO problem can be restructured into a quantum-computing-friendly Ising model format is adeptly explained in this paper [15].

The study that was mentioned earlier [6] utilized the QUBO/Ising model to run their experiments on a D-Wave 2000Q processor. Another study [16] performed on the applicability of Boolean satisfiability via quantum computing using the QUBO/Ising model simulated further enhancements to the D-Wave processors. The architecture routing and placement optimizations that were implemented demonstrated that larger problems could still be solved, with some extra runtime used to optimize placement and routing.

Current improvements that are being developed for QUBO on classical computers have also been tested on quantum annealers in recent studies [17]. The improvement in the mentioned paper was a novel way of estimating the density of states in Boolean satisfiability problems. It should be noted that the solutions to the problems that were benchmarked against had numerical solutions rather than Boolean ones; therefore, the solutions were rated on closeness rather than correctness. The two classical solvers that were used for the comparison with the D-Wave 2X processor were the Hamze-de Freitas-Selby (HFS) algorithm and satisfiability modulo theory (SMT) solvers. While the usage of the quantum annealer gave a marginal improvement over the HFS results, its results showed no significant improvements over the results yielded from the SMT solver.

Another approach was applied to QUBO/D-Wave optimization in another study [18]. This study was more application-oriented, as it tested its proposed enhancements when solving the Boolean multivariate quadratic (MQ) problem using a quantum annealer. Three different techniques of embedding/encoding were applied to the MQ problem: direct, truncated, and penalty embedding. Table 2 provides a comprehensive summary of the related works discussed in this section.

## 4. Methodology

### 4.1. B-SAT Experiment

The experiment began by constructing a dimacs file in CNF. A dimacs file is a text file that is used to describe a Boolean satisfiability problem in various forms. A dimacs file is parameterized by the number of variables, AND gate logical operators, OR gate logical operators, and then a proportion of literals have a NOT gate randomly applied to them. In the context of our study, the number of AND and OR gates was translated into literals and clauses in the following way:(14)#ANDs=#Clauses−1
(15)$$\#\mathrm{ORs}=\sum_{clause}\left(\#\mathrm{Literals}-1\right)$$

The CNF equation was checked for duplicates. If there was one duplicate of the same clause, the random allocation of NOT gates phase would restart. If the duplicates were detected after 100 reconstructions, then this would permit the existence of duplicates of clauses. The classes of B-SAT problems were divided by the number of variables/qubits, denoted by *n*, used in the construction of the quantum circuit. A n=3 B-SAT circuit would have 3 qubits expressing 3 boolean variables, and so on. The number of dimacs files per configuration can be seen in Table 3. The pattern that the dimacs CNF constructor followed can be exemplified in Table 4.

The dimacs file was then translated into a quantum circuit, with the intended adjustments made. The constructed circuit was then sent to IBM’s quantum processors to be run according to the specified number of shots. The whole experiment was coded in Python and Qiskit, IBM’s quantum computing library. The stages of the experiment are visually summarized in Figure 5.

The quantum circuits representing the Boolean SAT digital logic were run on three of IBM’s quantum processors: Quito, Lagos, and Toronto. The processor specifications are listed in Table 5 and were taken according to the calibrations that were recorded on 7 April 2023. Each circuit was executed 10 times (with the specified number of shots), and at the end of each execution, the result with the highest occurrence was tested on the actual equation to determine whether it was a valid solution or not.

Qubit mapping was performed using a library called “Mapomatic” [19]. Mapomatic is a library of qubit mapping functions that was developed by a group of researchers at IBM Quantum. It should be noted that the qubit mapping algorithm is not deterministic, meaning that the mapping procedure would sometimes yield slightly different mapping layouts. That being said, the difference was very slight and rarely exceeded 10 in quantum circuit depth. All quantum circuits were transpiled with an optimization_level=3, which automatically applied a heavy circuit mapping optimization pass, hence a mapped circuit had a double layer of qubit mapping passes applied.

### 4.2. Shots Experiment

This experiment was a branch-out experiment motivated by the unexpected results observed in the experiment described in Section 4.1. Our experiment began by recording the error per Clifford (EPC) data of a single physical qubit, in one of IBM’s superconducting quantum processors, using Qiskit’s standard randomized benchmarking library functions [20]. After performing these single qubit experiments, a bundle of quantum circuits was constructed, using the same library, which ranged from a certain quantum circuit depth to another. Each of the 1-qubit data collection experiments had to be performed within a relatively short time window before each multi-qubit standardized random benchmarking experiment, otherwise a large disparity would be noticed. The 1-qubit data were used as in the expected values $E_{i}$ in Equation (Equation 7).

Given a specified seed value, the randomized circuits always generated the same “random” standard benchmarking circuit. The depth of each Clifford was gradually increased in the bundled multi-qubit experiments, resulting in a range of circuits rather than just a single circuit depth. This variation in Clifford gate length was required by Qiskit’s standard randomized benchmarking functions, to analyze the EPC’s spread by calculating the variance, while also mapping the divergent behavior demonstrated in the reduced chi-squared of the EPC behavior.

These same experiments were performed with different amounts of qubits in a circuit, which affected the random circuit depth produced. The main variable in our experiments, which the whole experiment was built around, was the number of shots performed in each multi-qubit benchmarking execution. A summary of the experimental procedure can be seen in Figure 6.

The experiment’s main objective was to examine the effect that the number of shots had, concerning the number of qubits under a circuit depth that would not likely result in random noise (49–156), and the effects of having a higher variance effect (as demonstrated in Figure 4). The time at which the experiments were performed after the calibration of the quantum processor was also recorded with each run; however, it is not mentioned in the results, due to its irrelevance to the analysis.

## 5. Results

### 5.1. B-SAT Experiment

#### 5.1.1. Three Qubits (n = 3)

Our first experiment in this subsection ran two configurations of the three variable/qubit B-SAT problem. All circuits were set to n = 3, meaning that the number of variables in the circuit was 3. All of the circuits were run with and without reapplying qubit mapping. Both executions were set to 1024 shots per circuit execution. The results can be seen in Figure 7. The “Success Score” marking the Y-axis indicates how well the top result of the 10 quantum computer executions fared; 0 means that none of the top measurements in the quantum circuit runs yielded correct answers, and 1 means that all of the top measurements yielded correct answers. Due to the very simple nature of the circuit, the results were mostly perfect, with few dips in fidelity. It is also noticeable that the qubit mapping optimization seemed to aggravate the drop in the success score.

There was one unexpected trend that came out of the n = 3 SAT experiments. The circuit depth of the unmapped quantum circuits yielded better results as the depth went above 30 s. This could be due to coherent errors, as they had a similar pattern of oscillating in magnitude as more operations were applied to a qubit. Generally, circuit depth is seen as something that should be minimized as much as possible, but in this case, it was shown to be a positive marker, as seen in Figure 8. Qubit mapping in the n = 3 SAT circuit did not improve the results, as can be seen in Figure 9. Qubit mapping also resulted in a higher average circuit depth.

#### 5.1.2. Four Qubits (n = 4)

Other than raising the number of variables to four, the goal of this subsection was to explore the effect that qubit mapping with differing numbers of shots would have on the quantum circuits. Another goal was to compare the performance of the Quito and Lagos quantum processors. After incrementing the number of variables to n = 4, a very noticeable drop in fidelity was found, as marked in Figure 10, compared to Figure 7. It appears that applying qubit mapping and doubling the number of shots per execution did enhance the results, especially as the number of AND gates was increased. The standard number of executions was set to 1024 shots per circuit execution, this means that the “x2 shots” circuits had a set number of shots of 2048.

Surprisingly, executing the same Boolean satisfiability circuits on a quantum processor that had more qubits (Lagos) resulted in a lower “Success Score” than when executed on a quantum processor with fewer qubits (Quito); even though, in general, Lagos has a lower read-out and CNOT error rate, as outlined in Table 5. As marked in Figure 10, qubit mapping and the doubling of shots, when running the circuits on Quito, made for better results.

Using qubit mapping and doubling the number of shots was not as useful with Lagos as with Quito (as seen in Figure 11. While both qubit mapping and doubling the number of shots did indeed yield better results on average when the number of AND gates/clauses was increased, the number of 0 scores also increased with the Lagos processor. Doubling the number of shots without qubit mapping has gave us worse results.

The n = 4 SAT Quito circuit depth analysis revealed how circuit depth in this setting can negatively affect circuit fidelity, as shown in Figure 12. The standard n = 4 SAT Quito depth analysis revealed a very similar trend, despite having half (on average) of the circuit depth of the mapped implementation, while having noticeably worse results. The depth aspect in the Lagos implementations was less relevant and did not show any significant trend, other than a slight increase in correctness when the depth was at its highest (shown in Figure 13). This was most likely due to the increase in OR gates, which raised the random probability of success.

#### 5.1.3. Five Qubits (n = 5)

The number of shots here was adjusted to 4096 in all executions. As seen in Figure 14, both cases had a success rate that resembled the probability of random chances of success. However, we were surprised to find a particular trend when analyzing the success score against the quantum circuit depth. Figure 15 shows the polynomial of order 6 aggregate line of the standard n = 5 SAT Quito experiments. Quantum circuits with a depth over 1000 were shown to have a score approaching 0.9. The effect was also applicable to lower polynomial aggregation lines. This prompted us to cross-examine this with the random chance of success in Section 5.1.5.

We performed n = 5 experiments on another quantum processor, Kolkata, using IBM’s updated runtime library. Unfortunately, the results remained the same. The experiments were repeated while setting the resilience level to 1, which raised the chances of success by only 4–5%. To verify our modus operandi, we also ran some of the lower depth n = 5 circuit experiments on IBM’s mock quantum processor “Fake Kolkata”, which gave us almost perfect results (these mock processors have some simulated noise applied to them).

#### 5.1.4. Six Qubits (n = 6)

For the n = 6 SAT problem, we used IBM’s Toronto quantum processor, to examine if it could keep up with the increasing amounts of qubits required. The number of AND gates in these quantum circuit runs had no clear pattern (as seen in Figure 16), other than they were approximately at a level with the random noise success rate: 46% (more details in the Section 5.1.5). The number of shots run on Toronto was adjusted to 8192 to compensate for the increased size of the state vector. No comparison experiment was executed other than the standard one for the n = 6 SAT.

The depth surprised us again, by giving us an almost linear increase in success, as seen in Figure 17. It is vital to note, however, that the quantum circuit depth seemed to have an almost linear correlation to the random chance success score. This could be partly because circuits with a higher OR:AND gate ratio tend to have a higher probability of getting higher “Success Scores”, due to their higher span of satisfiable solutions. The probability of a random success answer will be discussed in more detail in the following subsection.

#### 5.1.5. Further Analysis

This subsection will discuss points that have not been brought up in the previous subsections and the possible reasoning behind the results. The vast majority of CNF equations have more than one valid solution, and this section aims to examine this span of valid/satisfiable answers. According to the n = 5 SAT and n = 6 SAT circuits that were generated for this study, this span is very predictable. As seen in Figure 18, the ratio of OR:AND gates is what decides how big the span of satisfiable solutions is. As the number of AND gates increases, the ratio becomes lower, thus pulling the success score down. This effect can be partially seen in the n = 5 SAT executions on Quito (Figure 14). The effect is also noticeable on the n = 4 Lagos graph in Figure 11), but not the n = 4 Quito graph in Figure 10.

When comparing the general experimental results with the probabilistic rates (Table 6), the picture becomes clearer. Modern-day quantum computers can handle n = 3 SAT and n = 4 SAT problems adequately enough, but when increasing n to 5 and above, they start to yield random noise results. The n = 4 SAT results were only significantly higher than the probabilistic trend in the case of the Quito (mapped + x2 shots). The rest of the n = 4 SAT implementations were close to the probabilistic stats in terms of average, median, and correlation. Trying to solve the problem on a larger quantum processor such as Lagos or Toronto did not give us better results, even though their median gate and readout error rates (see Table 5) were less than the smaller Quito processor.

Moreover, the correlations were quite clear (see Table 6) regarding how much the results were affected by the random probability’s safety umbrella. It should be noted that circuits with more OR gates resulted in a larger quantum circuit depth, while the number of AND gates did not have a noticeable effect on the depth. The proportion of NOT gates in the circuits did not have any significant effect on the results, hence not being mentioned in the previous subsections.

We had no conclusive evidence to explain the cause of the larger variance in the n = 4 circuits, especially in Lagos, and this led us to email IBM Quantum regarding this matter. They explained that the cause of this behavior is due to a miscommunication on whether more shots are coming, and that this behavior is both hardware and firmware version-specific. We were also assured by IBM Quantum that the issue should be patched soon in the next backend upgrade. For more information regarding the effect that the number of shots imposes, please refer to the results of the shots experiment (Section 5.2).

### 5.2. Shots Experiment

The results proved what we suspected from the B-SAT experiment, which is that the number of shots did affect the results in unusual ways. To be precise, the variance and fidelity did not peak at the lowest number of shots. As seen in Figure 19, the $\chi^{2}$ on the Kolkata quantum processor showed a predictable pattern by peaking at particular numbers of shots as the number of qubits was increased. When the number of qubits was five, the $\chi^{2}$ peaked at 2000 shots, and when the number of qubits was increased to six, the $\chi^{2}$ peaked at 4000 shots. The pattern persists with the 7-qubit circuits, where the $\chi^{2}$ peaked at 6000 shots. The 8-qubit circuit verified the previous pattern by $\chi^{2}$ showing a spike in $\chi^{2}$ at the 8000-shot mark. It also showed signs of obeying the conceptual rules demonstrated in Section 2.4, by having a higher variance when the number of shots was set to the lowest setting in our experiments.

Kolkata also showed these spikes in variance, but they peaked at a different number of shots than the $\chi^{2}$, as seen in Figure 19. The spike in variance seemed to be set at 6000 shots when the number of qubits in the circuits was seven and eight. Otherwise, the spike in variance followed the $\chi^{2}$ pattern when the number of qubits is five.

The other quantum processor, Cairo, displayed a similar spike in variance and $\chi^{2}$, which repeated with the same number of shots and qubits (as seen in Figure 20). This led us to speculate that this occurrence was not tied to a single quantum processor. The spike was also present in the Hanoi quantum processor, albeit at the 4000 shots mark rather than 2000 shots when running the 5-qubit circuit (shown in Figure 20).

## 6. Conclusions

After performing numerous experiments, we reached several key points regarding solving the B-SAT problem using quantum computing:Our results showed that the performance in solving B-SAT problems did not directly correlate with the size of the quantum processor, in terms of quantum volume or the number of qubits. Instead, factors such as implementing an extra procedure of qubit mapping and using a smaller processor were more indicative of successful outcomes. For example, in our experiments, the smaller Quito processor with higher median readout and CNOT error outperformed the larger Lagos processor when the number of variables in the B-SAT problem was set to four.Implementing an additional layer of qubit mapping consistently led to improved results. This was most noticeable in larger circuits, where qubit mapping helped mitigate error accumulation and better utilize the hardware’s limited qubit connectivity. Although not a silver bullet for all noise issues, this technique proved to be an important optimization in improving the fidelity of our quantum circuit executions.The quantum processors performed reliably on the n = 3 SAT and n = 4 SAT circuits. However, for the n = 5 SAT and n = 6 SAT circuits, the accuracy of the results significantly dropped, approaching random outputs, due to the accumulation of quantum noise and errors. This suggests that as the number of qubits increases, noise and gate errors begin to overwhelm the system, resulting in outputs that are no longer distinguishable from random guesses, commonly referred to as reaching ”noise level”. This phenomenon highlights the current limitations of quantum processors when tackling larger SAT problems in the NISQ era.Increasing the number of shots did not result in a lower result variance, as it increased the variance in a certain higher number of shots. According to IBM, the cause of this issue was the firmware on their quantum processor. This hardware–firmware issue has affected numerous quantum experiments, not just our B-SAT experiment.A bigger circuit depth does not always imply worse results (at least not for an SAT circuit), as it sometimes gave us better results even when taking the random chance of success into account.

## Figures and Tables

**Figure 1 entropy-26-00875-f001:**
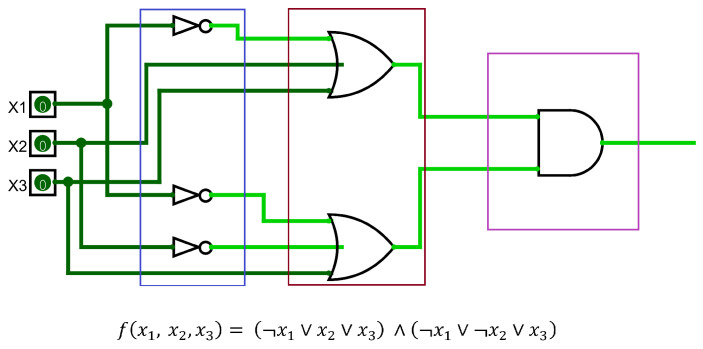
A CNF circuit represented in digital logic form, with its different stages outlined.

**Figure 2 entropy-26-00875-f002:**
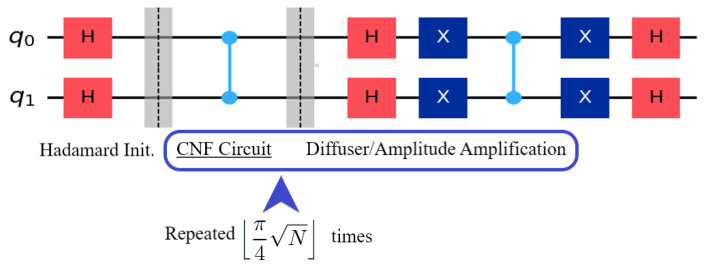
In this example, the CNF circuit would be $f\left(x_{1},\;x_{2}\right)=x_{1}\wedge x_{2}$. It should be noted that this example has only one distinct solution.

**Figure 3 entropy-26-00875-f003:**
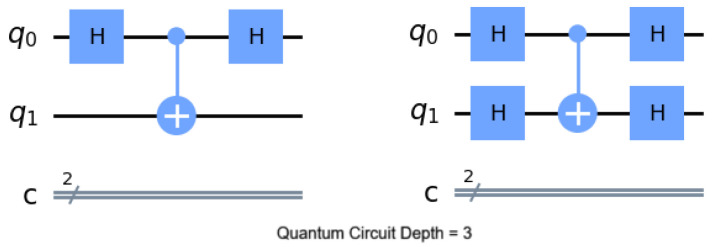
While these two quantum circuits have almost double the amount of quantum gates, they have an identical quantum circuit depth. Parallel operations do not accumulate.

**Figure 4 entropy-26-00875-f004:**
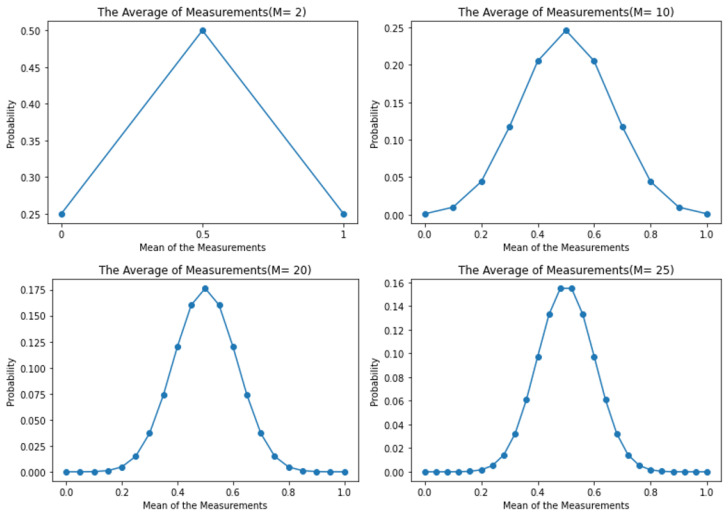
Variance of random classical variable vs. probability of measure 1. The probability of yielding outlier mean measurements decreases as the number of shots is increased, hence this should be an indicator that the variance should decrease as the number of shots is increased.

**Figure 5 entropy-26-00875-f005:**

A brief summarization of the experimental procedure.

**Figure 6 entropy-26-00875-f006:**
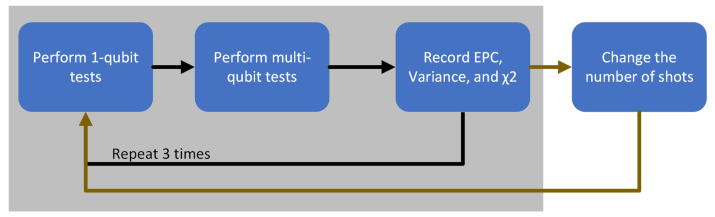
This is a figure that summarizes our experimental procedure. The black arrows were repeated 3 times per cycle, while the olive-colored lines were performed once per cycle.

**Figure 7 entropy-26-00875-f007:**
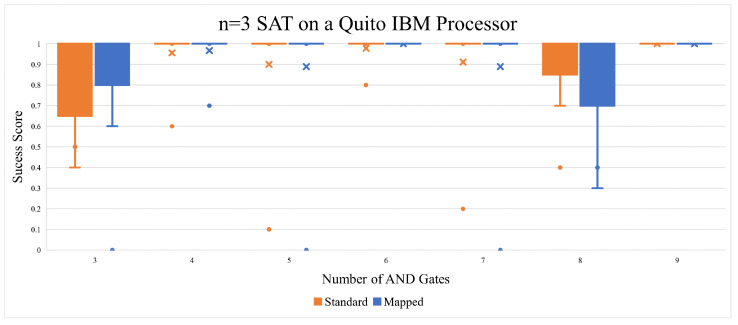
The n = 3 SAT results on Quito gave us almost perfect answers, with a few outliers here and there. The dots represent outlier results that fall outside the expected range. Specifically, they are points significantly higher or lower than the rest of the data. The x mark in the box plot often indicates the mean (average) value of the dataset.

**Figure 8 entropy-26-00875-f008:**
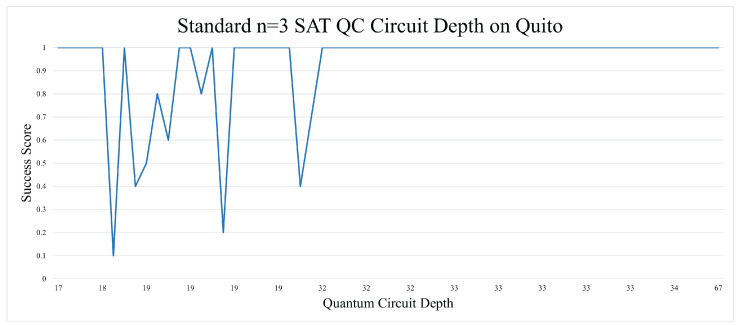
Circuit depth seemed to be tied to more positive results with the unmapped n = 3 B-SAT.

**Figure 9 entropy-26-00875-f009:**
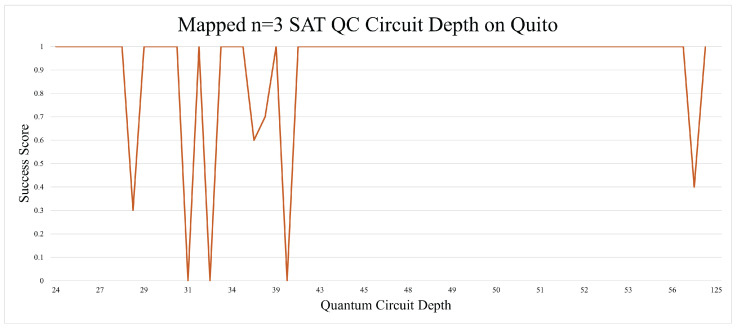
Qubit mapping did not improve the n = 3 SAT results by much.

**Figure 10 entropy-26-00875-f010:**
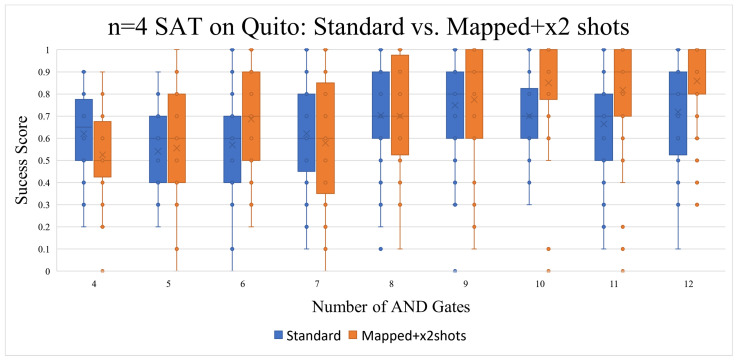
Increasing the n to 4 marked a noticeable decline in the success score, but the results showed some improvement when using qubit mapping and increasing the number of shots per execution.

**Figure 11 entropy-26-00875-f011:**
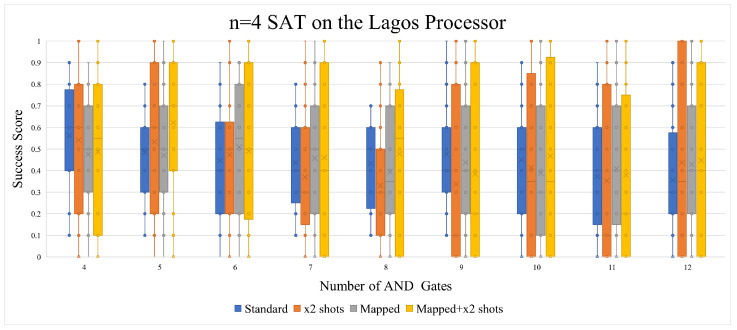
Either/or doubling the number of shots and using qubit mapping resulted in an aggravated success score.

**Figure 12 entropy-26-00875-f012:**
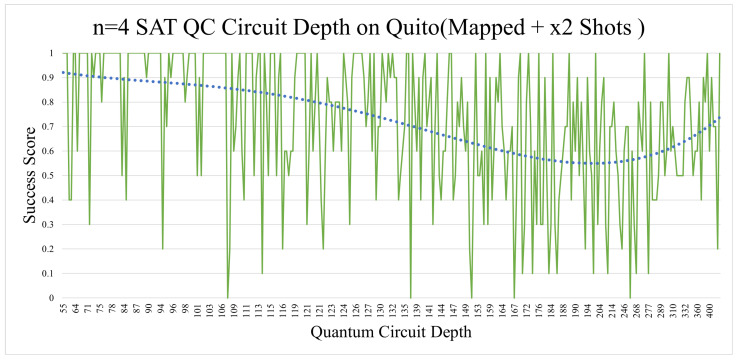
The dotted blue line marks a polynomial aggregation of the experimental results.

**Figure 13 entropy-26-00875-f013:**
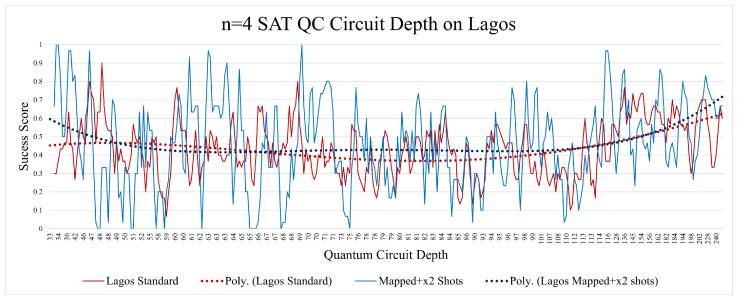
There was no noticeable pattern with the depth aspect of the n = 4 SAT runs on Lagos.

**Figure 14 entropy-26-00875-f014:**
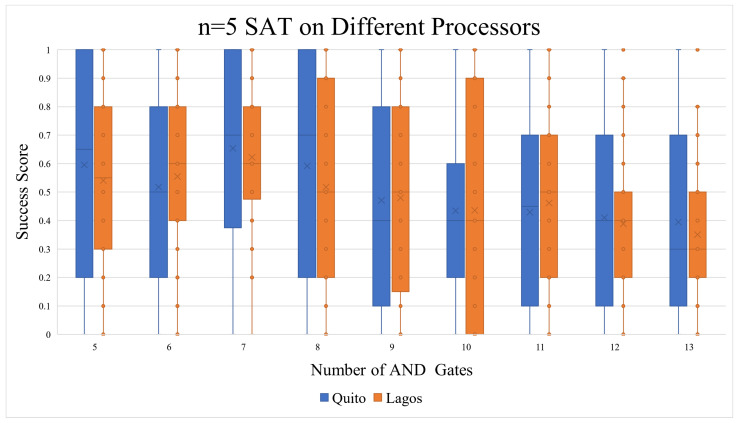
Both Quito and Lagos seemed to produce similar results, which would be interpreted as noise when factoring in the random chance of success.

**Figure 15 entropy-26-00875-f015:**
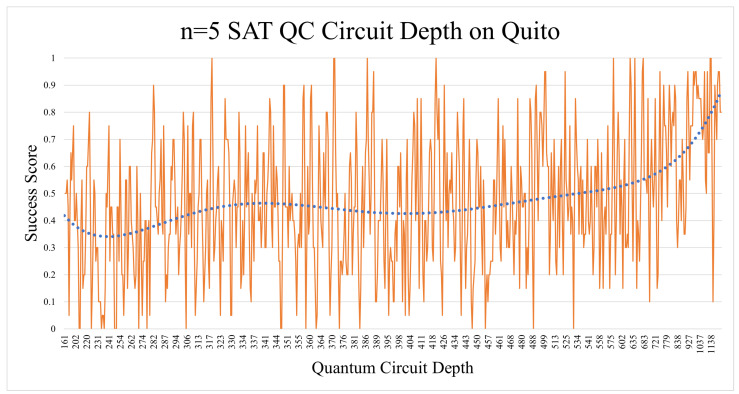
In this case, the n = 5 SAT experiments indicated that the high circuit depth was caused by a higher number of OR gates, which resulted in a higher probability of random success.

**Figure 16 entropy-26-00875-f016:**
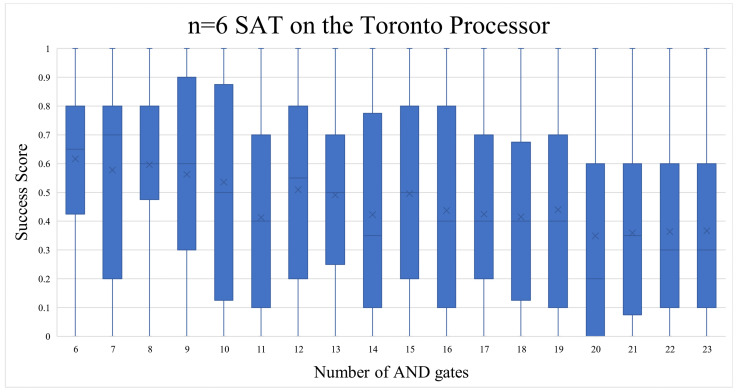
As the number of AND gates increased, the results dropped to just noise level integrity.

**Figure 17 entropy-26-00875-f017:**
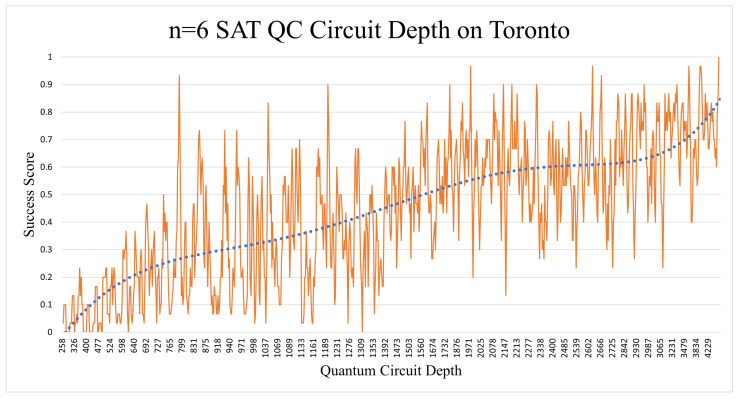
The correlation of a higher circuit depth and success score could have been caused by the circuits having a higher proportion of OR to AND gates.

**Figure 18 entropy-26-00875-f018:**
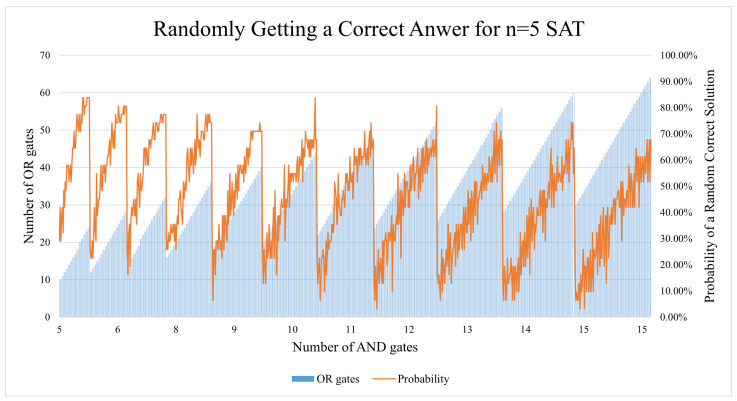
The ratio of OR:AND gates is what decides how large the span of satisfiable solutions is.

**Figure 19 entropy-26-00875-f019:**
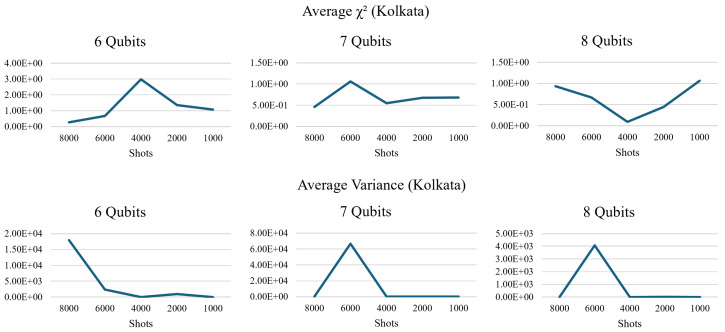
The experiments executed with differing numbers of qubits on Kolkata showed an atypical predictable fidelity loss spike, shifting to the left with each increment in the number of qubits (**upper row**). The variance spiked at 8000 and 6000 shots with the 5+ qubit circuits (**lower row**).

**Figure 20 entropy-26-00875-f020:**
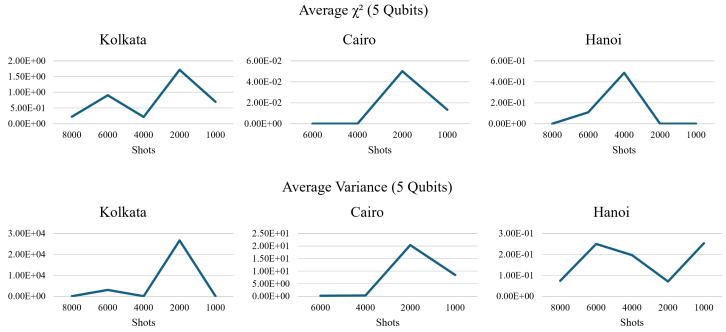
The number of shots across the five different quantum processors seemed to yield unusual results that did not follow the mathematical model (**lower row**). The drop in fidelity measured by the $\chi^{2}$ was also strangely tied to the number of qubits (**upper row**).

**Table 1 entropy-26-00875-t001:** An outline of the classical and quantum expectation value squared.

Classical $E[X^{2}]$	Quantum $E[X^{2}]$
$\begin{array}{ccc} E[X^{2}] & = & \sum_{x\in 0,1}x^{2}Pr\left[X=x\right]\\ & = & 0Pr[X=0]+1Pr[X=1]\\ & = & p \end{array}$	$\begin{array}{ccc} E[X^{2}] & = & \sum_{x\in 0,1}x^{2}Pr\left[X=x\right]\\ & = & \sum_{x\in 0,1}x^{2}\langle \hat{\mu }\left(x\right)\rangle \\ & = & \langle \sum_{x\in 0,1}x^{2}\hat{\mu }\left(x\right)\rangle \\ & = & \langle {\hat{M}}^{2}\rangle \end{array}$

**Table 2 entropy-26-00875-t002:** Related work summarized.

Paper Name	*Authors*	*Year*	*Model*	*Optimization*	*Execution/Simulation Platform*	*Qubits*
Quantum cooperative search algorithm for 3-SAT [10]	S. Cheng and M. Tao	2006	Grover Search	Variational GenSAT	Mathematically simulated (Mathematica Implied)	0–20 Simulated qubits
A Quantum Annealing Approach for Boolean Satisfiability Problem [16]	J. Su, T. Tu, and L. He	2016	QUBO/Ising	D-Wave architecture routing and placement optimization	Mathematically simulated	12 × 12 cell and 100 × 100 cell architecture
Assessing Solution Quality of 3SAT on a Quantum Annealing Platform [6]	T. Gabor et al.	2019	QUBO/Ising	Logical Postprocessing	D-Wave 2000Q System	2048 quantum annealing qubits
Estimating the Density of States of Boolean Satisfiability Problems on Classical and Quantum Computing Platforms [17]	T. Sahai, A. Mishra et al.	2020	QUBO/Ising	State density estimation of Boolean problems	D-Wave 2X System	1152 quantum annealing qubits
Finding Solutions to the Integer Case Constraint Satisfiability Problem Using Grover’s Algorithm [11]	G. M. Vinod and A. Shaji	2021	Grover Search	Adding thermal relaxation and and depolarization noises	Ibmq_qasm_simulator and ibmq_16_melbourne	Up to 32 simulated qubits and 14 UG qubits
Impact of Various IBM Quantum Architectures with Different Properties on Grover’s Algorithm [12]	M. H. Akmal Zulfaizal Fadillah et al.	2021	Grover Search	Qiskit parameter optimization	ibmq_16_santiago, ibmq_16_belem, ibmq_16_yorktown, ibmq_16_melbourne	5–14 UG Qubits
Solving Systems of Boolean Multivariate Equations with Quantum Annealing [18]	S. Ramos-Calderer et al.	2022	QUBO/Ising	Direct, truncated, and penalty embedding	D-Wave Advantage System	5760 quantum annealing qubits

**Table 3 entropy-26-00875-t003:** Number of dimacs files generated per configuration.

SAT Configuration	Number of Dimacs Files Generated
n = 3	64 files
n = 4	325 files
n = 5	709 files
n = 6	880 files

**Table 4 entropy-26-00875-t004:** A sample of the executed circuits in terms of parameters.

n = 3 B-SAT	3 AND gates	6 OR gates	30% NOT gate application
			50% NOT gate application
			70% NOT gate application
		7 OR gates	30% NOT gate application
			50% NOT gate application
			70% NOT gate application
		8 OR gates	30% NOT gate application
			50% NOT gate application
			70% NOT gate application
	4 AND gates	8 OR gates	30% NOT gate application
			50% NOT gate application
			70% NOT gate application
		9 OR gates	30% NOT gate application
			50% NOT gate application
			70% NOT gate application
		10 OR gates	30% NOT gate application
			50% NOT gate application
			70% NOT gate application

**Table 5 entropy-26-00875-t005:** Quantum processor specifications.

Processor	*Qubits *	*QV*	*Median Readout ERR*	*Median CNOT ERR*
Quito	5	16	4.250 × 10^−2^	1.012 × 10^−2^
Lagos	7	32	1.667 × 10^−2^	7.135 × 10^−3^
Toronto	27	32	1.910 × 10^−2^	1.009 × 10^−2^

**Table 6 entropy-26-00875-t006:** Statistical comparison between the experimental results and probabilistic chance of random success.

Quantum Processor	Probablistic
n = 3 SAT on Quito	n = 3 SAT
Average: 93%	Average: 39%
Median: 100%	Median: 43%
n = 3 SAT Correlation: −0.5311	
n = 4 SAT on Quito (Mapped + x2 Shots)	n = 4 SAT
Average: 73%	Average: 42%
Median: 80%	Median: 40%
n = 4 SAT Correlation: −0.4132	
n = 5 SAT on Quito	n = 5 SAT
Average: 50%	Average: 46%
Median: 50%	Median: 48%
n = 5 SAT Correlation: 0.4852	
n = 6 SAT on Toronto	n = 6 SAT
Average: 44%	Average: 44%
Median: 40%	Median: 45%
n = 6 SAT Correlation: 0.6176	

## Data Availability

The source code used to conduct the experiments in this study is available at https://github.com/aabennak/Bool-Sat.git (accessed on 16 October 2024). The repository contains the source code for the B-SAT experiment and the shots experiment.

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
