# Peer review of "Solving the B-SAT Problem Using Quantum Computing: Smaller Is Sometimes Better"

_entropy, 2024, doi:10.3390/e26100875_

Round 1
Reviewer 1 Report
Comments and Suggestions for Authors
The paper provides a series of benchmarks executed using SAT problem. The Authors provide a details study of the behaviour of IBM machines used for the purpose of solving an instance of this problem.
In my opinion, the presented paper can be potentially published in Entropy. However, the major weakness of the presented paper is the conclusions section. The authors should significantly improve this section, in particular by addressing the following issues.
- Some conclusions are very vague. Firstly, "The size, in terms of quantum volume and qubits, of a quantum processor is not an indicator of how well it performed in our B-SAT experiments." is not a conclusion. If not the volume and qubits, then what? Also, I don't understand how the comment 'The extra layer of qubit mapping almost always resulted in improved average results' is supposed to be helpful for the reader?
- What does it mean that 'the results declined to noise level for the n=5 SAT and n=6 SAT."?
- I also strongly recommend consulting the recently published preprint https://arxiv.org/abs/2406.11771 It would be beneficial if the authors could provide some remarks concerning the results presented in this preprint,
Additionally, as the paper is based on the numerical experiments, the full source code, along with the specification of experiments and detailed instructions on how to reproduce the presented results, should be provided.
The paper could be published after providing a detailed response to the above remarks.
Author Response
Comments 1: [In my opinion, the presented paper can be potentially published in Entropy. However, the major weakness of the presented paper is the conclusions section. The authors should significantly improve this section, in particular by addressing the following issues.]
Response 1:[Thank you for pointing this out, we have made adjustments to make our conclusion points easier to understand. We can see now on hindsight that they could have been laid out in a better way.]
Comments 2: [I also strongly recommend consulting the recently published preprint https://arxiv.org/abs/2406.11771 It would be beneficial if the authors could provide some remarks concerning the results presented in this preprint,]
Response 2: [Thank you again for referring us to this paper. It was very informative to look at to structure future quantum computing studies. However, our submitted study's experiments have all been done in the pre-Qiskit 1.x versions so any additional experimentation would present an unspecified number of controls changed. ]
Comments 3: [Additionally, as the paper is based on the numerical experiments, the full source code, along with the specification of experiments and detailed instructions on how to reproduce the presented results, should be provided.]
Response 3: [The source code of the experiments has been added in the data availability section.]

Reviewer 2 Report
Comments and Suggestions for Authors
Review of "Solving the B-SAT problem using quantum computing: Smaller is Sometimes Better"
This paper gives a detailed account of solving the B-SAT problem through a standard quantum search algorithm. The authors systematically vary parameters within the approaches, in particular the choice of actual quantum hardware, the number of qubits used in the encoding and the number of shots.
The results are detailed and represent an interesting snapshot of the possibility of solving this somewhat archetypal problem on current noisy quantum hardware. The conclusions provide some useful rules of thumb for authors attempting similar problems.
Overall, the paper is written well. Much of the text in the figures is too small - much smaller than the main text of the paper, and is hard to read, which should be fixed. Otherwise, I find the manuscript suitable for Entropy.
Author Response
Comments 1: [Overall, the paper is written well. Much of the text in the figures is too small - much smaller than the main text of the paper, and is hard to read, which should be fixed. Otherwise, I find the manuscript suitable for Entropy.]
Response 1: [Thank you for reading the paper. Can you please kindly specify the figures that need their fonts size increased?]